# The speed limit of optoelectronics

M. Ossiander [1,6 ✉], K. Golyari[1,2], K. Scharl[1,2], L. Lehnert[1,2], F. Siegrist[1,2], J. P. Bürger[1,2], D. Zimin[1,2], J. A. Gessner[1,2], M. Weidman[1,2], I. Floss[3], V. Smejkal[3], S. Donsa [3], C. Lemell [3], F. Libisch [3], N. Karpowicz [4], J. Burgdörfer[3], F. Krausz [1,2 ✉] & M. Schultze [2,5]

Light-field driven charge motion links semiconductor technology to electric fields with attosecond temporal control. Motivated by ultimate-speed electron-based signal processing, strong-field excitation has been identified viable for the ultrafast manipulation of a solid's electronic properties but found to evoke perplexing post-excitation dynamics. Here, we report on single-photon-populating the conduction band of a wide-gap dielectric within approximately one femtosecond. We control the subsequent Bloch wavepacket motion with the electric field of visible light. The resulting current allows sampling optical fields and tracking charge motion driven by optical signals. Our approach utilizes a large fraction of the conduction-band bandwidth to maximize operating speed. We identify population transfer to adjacent bands and the associated group velocity inversion as the mechanism ultimately limiting how fast electric currents can be controlled in solids. Our results imply a fundamental limit for classical signal processing and suggest the feasibility of solid-state optoelectronics up to 1 PHz frequency.

[1] Max-Planck-Institut für Quantenoptik, Hans-Kopfermann-Str. 1, 85748 Garching, EU, Germany. [2] Fakultät für Physik, Ludwig-Maximilians-Universität München, Am Coulombwall 1, 85748 Garching, EU, Germany. [3] Institute for Theoretical Physics, Vienna University of Technology, Wiedner Hauptstrasse 8-10, 1040 Vienna, EU, Austria. [4] CNR NANOTEC Institute of Nanotechnology, via Monteroni, 73100 Lecce, EU, Italy. [5] Institute of Experimental Physics, Graz University of Technology, Petersgasse 16, 8010 Graz, EU, Austria. [6] Present address: John A. Paulson School of Engineering and Applied Sciences, Harvard University, 29 Oxford St, Cambridge, MA 02138, USA. ✉email: mossiander@g.harvard.edu; ferenc.krausz@mpq.mpg.de

The ability to control optical fields[1–3] has opened the door for exploring the ultimate rapidity at which electronic signals can be processed in circuitry. For advancing contemporary electronics toward optical clock rates (0.3–1 PHz), the laws that govern the creation of Bloch wavepackets, their acceleration, motion, interaction, and finally their coupling to optoelectronic interconnects need to be established.

Previously, strong-field excitation by ultrashort pulsed light has permitted wavepacket creation within sub-femtosecond time intervals as an ultrafast analog to the source signal in a field-effect transistor[4–16], opening a pathway to optical-rate signal processing. The method populates several conduction bands with an occupation profile quickly decreasing with energy (Fig. 1a). This hallmark of strong-field excitation implies two major drawbacks for high-fidelity and high-speed signal processing: (a) a reduced effective bandwidth available to sustain short signal transients and (b) multi-band electron wavepackets prohibiting reliable mapping of optical to electronic signals and their subsequent coupling to external circuitry.

For avoiding these shortcomings, we create a one-femtosecond single-band Bloch wavepacket via single-photon excitation to exploit the full width of individual energy bands homogeneously and selectively. Its subsequent motion is steered by a few-cycle optical "gate" field and interrogated by the current the gate-field-induced charge separation causes in an external circuit. We have dubbed this approach linear petahertz photoconductive sampling (LPPS). Being the sub-femtosecond-resolution analog to an Auston switch[17,18], LPPS enables the real-time probing of electronic processes defining the frontiers of optoelectronics.

To explore the limits of classical-field-driven electronics, we selected the widest-bandgap naturally abundant dielectric, lithium-fluoride (LiF, bandgap $E_{\text{Gap}} = 13.6\,\text{eV}$[19], width of the first conduction band $\Delta E_{\text{CB1}} = 6.2\,\text{eV}$[20]) for our studies. Time-resolved optical-field-induced charge currents along with theoretical modeling based on the optical Bloch equations[21] reveal

how population transfer to higher conduction bands impedes high-fidelity optoelectronic signal control and thereby sets a limitation for the processing rate attainable in future optoelectronic devices.

## Results

**Experimental concept.** Experimentally, we generate ultrashort vacuum-ultraviolet (VUV) light pulses (henceforth source pulses) by low-order harmonic generation in argon gas injected into high vacuum and exposed to controlled near-single-cycle laser waveforms (see Fig. 1c and the "Methods" section). The interaction occurs at the interface of perturbative and non-perturbative nonlinear optics. The VUV pulses with a duration on the order of 1 fs (discussed below) are tuned to largely confine carrier injection from the valence band to the first conduction band of LiF (CB1) via one-photon excitation (Fig. 1a, see the "Methods" section, and refs. [22–25]). The resultant carrier distribution $\rho_0(n, \boldsymbol{k})$ in band $n$ and spread over crystal momenta $\boldsymbol{k}$ of the Brillouin zone inherits the temporal and spectral characteristics of the VUV pulses, only slightly perturbed by the slowly varying amplitude and phase distribution of the transition dipole moment over the spectral range of interest[26]. The source pulse generates no initial current due to the momentum-symmetric carrier distribution $\rho_0(n, \boldsymbol{k}) = \rho_0(n, -\boldsymbol{k})$.

The electron–hole pairs are then separated by the electric field of a visible-to-near-infrared laser pulse (henceforth referred to as the gate field/pulse) delayed with respect to the source pulse with sub-femtosecond precision (by the time-shift $\tau$). The interaction region of source and gate field in the LiF sample is surrounded by two metal electrodes deposited on its surface. The dipole formed by the gate-field-induced electron–hole separation drives a current through the external circuit connecting the two electrodes[27,28] (details in the "Methods" section) to shield the dipole field in the electrodes (depicted by the mirror charges in

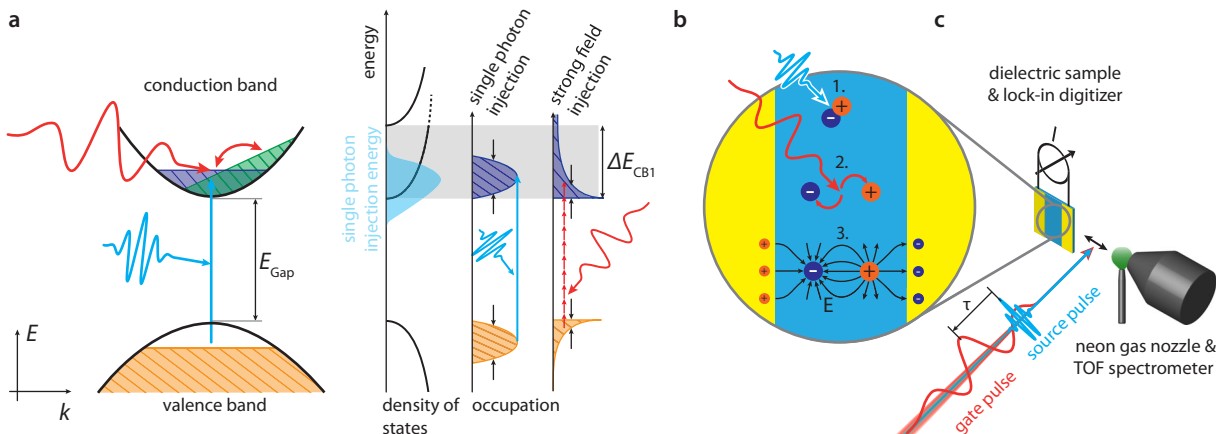

**Fig. 1 Linear petahertz photoconductive sampling (LPPS) in dielectrics. a** Left Panel: momentum-space picture (energy $E$ versus crystal momentum $k$): vacuum-ultraviolet (VUV) radiation (light blue) excites electrons across the bandgap $E_{\text{Gap}}$ from the valence (orange area) to the lowest conduction band CB1 (dark blue area) with conduction band width $\Delta E_{\text{CB1}}$ where the gate waveform (red) modulates their momentum. An asymmetric wavepacket distribution (green area) after the duration of the gate pulse leads to a detectable current. Right panel: single-photon injection allows addressing specific regions of the band structure and controlling the spectral shape of the injected Bloch wavepacket via tuning of the photon energy. Thus, single-photon injection can avoid populating multiple conduction bands while at the same time injecting a spectrally broad (black arrows) wavepacket. Contrary, the strong field injection process dictates the spectral shape of the injected wavepacket and will either populate multiple conduction bands or generate a wavepacket with narrow effective bandwidth. **b** Real-space picture: the VUV radiation creates electron–hole pairs (dark blue and orange) in lithium fluoride (LiF, light blue) which are separated by the gate laser electric field. The resulting dipole induces image charges in electrodes (yellow) which are detected as a current. **c** Experimental setup: carrier-envelope phase-stable visible few-cycle laser pulses create short VUV bursts via nonlinear up-conversion in an argon gas target (not shown) under vacuum conditions. We focus the VUV source pulses and the visible gate fields (time-shifted by $\tau$, $\tau < 0$: delayed, $\tau > 0$: advanced) onto dielectric LiF (blue) and record the delay-dependent current signal $I$ along the gate-light polarization via two electrodes (yellow). The detected gate laser electric field can be benchmarked in-situ by replacing LiF with a gas nozzle (green) and TOF spectrometer for attosecond streaking.

Fig. 1b). Measuring this current yields information about the magnitude of the dipole induced by the gate field.

**Microscopic origin of the current.** According to Bloch's acceleration theorem[29] and in the absence of scattering, the crystal momenta of the charge carriers are shifted by the instantaneous vector potential in the Coulomb gauge $A(t) = \int_t^\infty E_{\mathrm{Gate}}(t')\mathrm{d}t'$ of the external (gate) electric field, $E_{\mathrm{Gate}}$. Consequently, carriers injected into the conduction band at time $\tau$ with crystal momentum $k_0$ will shift to the asymptotic $k_{\mathrm{final}}$ after the end of the gate pulse ($t \to \infty$):

$$k_{\mathrm{final}}(\tau) = k_0 + \frac{1}{c}A(\tau) \tag{1}$$

The sequence of the source and the gate pulses creates a final carrier distribution $\rho_{\mathrm{final}}(n, k, \tau)$ that depends on the time delay $\tau$ between the pulses. Without transitions between conduction bands, $\rho_{\mathrm{final}}(n, k, \tau) = \rho_0\left(n, k - \frac{1}{c}A(\tau)\right)$. Thus, the incident electric field is converted to a current density $j(\tau)$ which is proportional to the average group velocity of the final carrier distribution. Introducing the band and momentum-dependent group velocity $v_g(n, k) = \frac{1}{\hbar}\nabla_k \varepsilon(n, k)$ that is determined by the dispersion relation $\varepsilon(n, k)$ of the crystal yields

$$j(\tau) \propto \sum_n \int_{\mathrm{BZ}} \rho_{\mathrm{final}}(n, k, \tau)\, v_g(n, k)\, \mathrm{d}k. \tag{2}$$

In regions of the band structure with nearly quadratic dispersion relation $\varepsilon(n, k) \propto k^2$, in the present case near the bottom of CB1, the final group velocity depends linearly on the applied vector potential for all occupied $k$. This corresponds to a Bloch wavepacket that is non-dispersive in momentum space. The red trace shown in Fig. 2a depicts the charge flowing through an electrometer in the external circuit induced by $j(\tau)$. Since the current $j(\tau)$ is caused by the momentum shift and directly proportional to the instantaneous vector potential $A(\tau)$, we expect the measured LPPS signal (charge), $S(\tau)$, to map out the gate field vector potential.

**Gate field measurement.** We verify this conjecture by concurrently determining the gate vector potential $A(t)$ from an in situ attosecond streaking measurement (Fig. 2b and see the "Methods" section), the tried and tested[30,31] method for optical field sampling. The comparison between the LPPS signal and $A(t)$ (Fig. 2a) confirms the linear relationship between $S(\tau)$ and the vector potential to be recorded. This establishes LPPS as a high-fidelity, rapid acquisition technique for petahertz-scale optical field retrieval without the need for photoelectron energy analysis in high vacuum[32]. In our current setup, we achieve linear operation up to a gate peak intensity of 1.7 V/nm, owing to the large bandgap and the large conduction band bandwidth of LiF.

**VUV pulse measurement.** The traces displayed in Fig. 2a, c represent a cross-correlation between the temporal profile of electron-hole pair generation and the gate waveform[33], the latter precisely known from attosecond streaking (see the "Methods" section). Hence, deconvolution allows determining the duration of the injection time window, $\Delta\tau_{\mathrm{FWHM}} \approx 1.4$ fs. This demonstrates switching the conductivity of a dielectric on the one-femtosecond time scale by linear absorption, i.e. an Auston switch[18] with near-petahertz bandwidth.

In analogy to attosecond streaking[30,31,34], the injection pulse duration should be comparable to or shorter than a half cycle of the gate field. In our experiment, the duration of carrier injection imposes a cut-off frequency of 0.4 PHz (wavelength ≈ 710 nm). The gate signal is transferred into an electric current with high

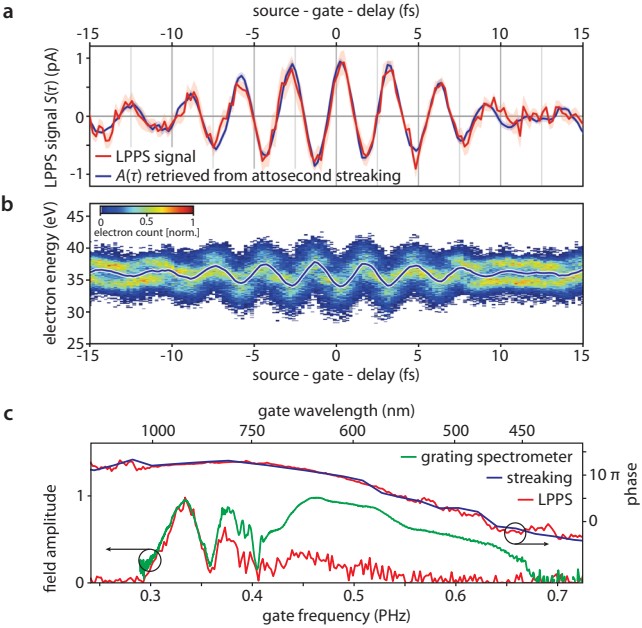

**Fig. 2 Optical field sampling using linear petahertz photoconductive sampling (LPPS). a** LPPS signal $S(\tau)$ (red line, standard deviation red-shaded area) as a function of the source-gate-delay (the source pulse precedes the gate pulse for negative delays) at a gate intensity of 1.7 V/nm compared to the vector potential of the gate laser electric field retrieved via attosecond streaking $A(\tau)$ (blue, standard deviation blue-shaded area). **b** Attosecond streaking spectrogram (false color plot) and extracted steaking momentum shift (solid line). The streaking momentum shift is linearly proportional to the negative vector potential of the gate laser pulses (blue line in panel (**a**)). **c** Spectral amplitude and phase of the electric field of the gate laser pulses retrieved via differentiation of the vector potential retrieved via LPPS (red) compared to the spectral phase retrieved via attosecond streaking (blue) and the gate field spectral amplitude measured with a calibrated grating spectrometer (green). See the "Methods" section and Supplementary Fig. 1 for details.

fidelity up to this frequency. Beyond this limit, the spectral sensitivity rolls off (Fig. 2c). Spectral correction should allow measuring signals beyond 0.5 PHz in our current setting due to the accurate phase retrieval above 0.5 PHz, revealed by Fig. 2c. The lifetime of photoinjected carriers in LiF has previously been determined to be on the order of 0.5 ps[35], setting an upper limit to the maximum gate pulse duration.

**Observation of conduction band electron dynamics.** The induced charge-carrier dynamics and the limits on the fidelity and on the linearity of ultrafast wavepacket steering can be probed by inspecting $S(\tau)$ for increasing gate field intensity. Upon raising the gate power incident on LiF we find an increasing difference $\Delta S(\tau)$ between the recorded LPPS signals and a scaled low-intensity reference recording (see Fig. 3b, c and see the "Methods" section). By contrast, $\Delta S(\tau)$ is not noticeable even at the highest gate amplitudes in a control experiment with a neon gas target between two metallic electrodes in the LPPS setup (Fig. 3a, see the "Methods" section for details).

An expected source of $\Delta S(\tau)$ in LiF is the flattening of the conduction band dispersion near the zone boundary and the resulting deviation from the free-particle behavior $\varepsilon(\mathrm{CB1}, k) \propto k^2$. The strength of this intraband deviation depends only on the final crystal momentum and thus is proportional to the amplitude of the gate field vector potential $A(\tau)$ at the moment of carrier injection (Eq. (1)). Therefore, the effect monotonically increases

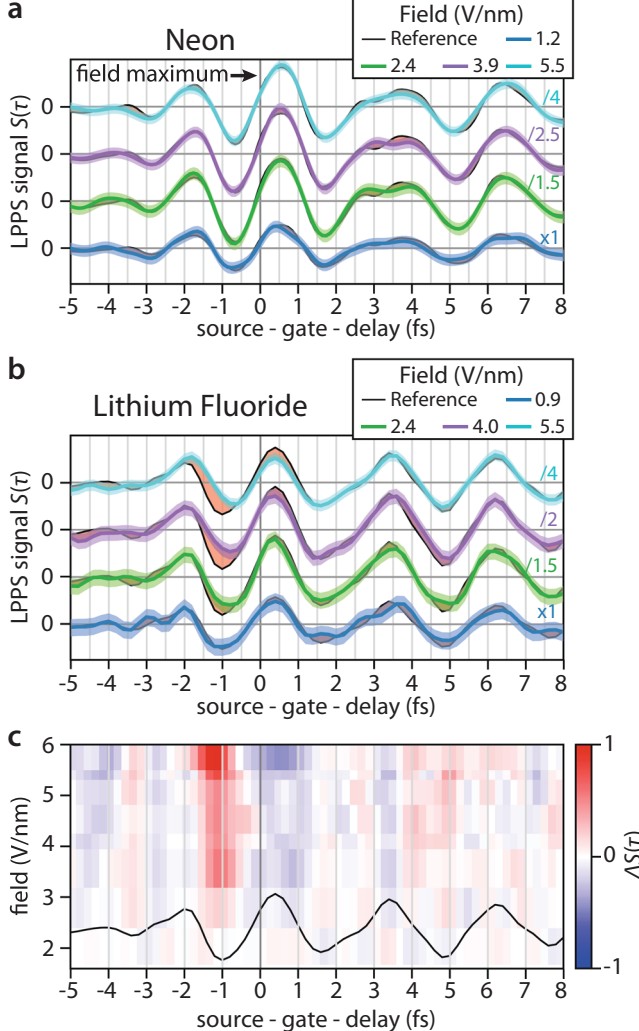

**Fig. 3 Gate field intensity-dependent linear petahertz photoconductive sampling (LPPS).** LPPS measurements in **a** Ne and **b** LiF. We show LPPS signal $S(\tau)$ traces (colored) versus a reference (black) recorded at 1.7 V/nm peak gate intensity. Color-shaded areas (except red) denote standard deviations including the reference noise. At low gate field amplitudes (≤2.4 V/nm), the LPPS signal deviation $\Delta S(\tau)$ is within the measurement error bar, whereas significant deviations (marked red-shaded) appear before the gate pulse maximum at peak intensities higher than 2.4 V/nm. These measurements were taken on different days compared to the measurement in Fig. 2, therefore the detected gate field vector potentials are different. **c** Time- and field-resolved LPPS signal deviations $\Delta S(\tau)$, black line: reference gate field temporal evolution. For clarity, signals are constrained to the gate field frequency spectrum below 1.2 PHz.

with the measured signal, can be corrected for in post-processing, and levels off the vector potential extrema symmetrically under the gate field intensity envelope. However, we observe the largest deviations $\Delta S(\tau)$ one optical half-cycle prior to reaching the gate maximum (Fig. 3c). This shift implies that $\Delta S(\tau)$ is dominated by effects occurring about one to two femtoseconds after carrier injection. Our analysis in the next paragraphs reveals that the magnitude and timing of the deviation $\Delta S(\tau)$ provide time-resolved insights into the multi-band electron dynamics and allow to distinguish between interband and intraband effects of the wavepacket evolution.

After carrier excitation by the source pulse (blue arrow in Fig. 4a), the wavepacket is driven through the Brillouin zone in

CB1. When the wavepacket reaches regions of the Brillouin zone with large coupling strength to the second conduction band CB2, it is split into two parts by gate field-driven non-adiabatic Landau–Zener transitions[36,37] (red arrow in Fig. 4a). After the gate pulse has terminated, the split parts of the wavepacket still possess the same final crystal momentum $\boldsymbol{k}_0 + \boldsymbol{A}(\tau)/c$. However, they are distributed over multiple bands that feature inverted group velocities, and therefore the LPPS signal is strongly suppressed. As Landau–Zener transitions to higher conduction bands occur only after intraband transport to the avoided crossings in the band structure, the deviation $\Delta S(\tau)$ is most pronounced for negative source-gate-delays, i.e., when the VUV carrier injection precedes the peak of the gate field.

**Theoretical modeling and verification.** We verify the wave-packet dynamics by solving the time-dependent optical Bloch equations[38,39]. They incorporate the band structure and dipole-coupling matrix elements for LiF along the Γ–X direction deduced from density-functional theory (see the "Methods" section). While quantitative agreement between simulation and experiment cannot be expected from models that do not account for the entire three-dimensional Brillouin zone as the one used here, they may still provide qualitative insight into the processes involved in the LPPS signal generation within the solid. As in the experiment, the simulated current follows $\boldsymbol{A}(\tau)$ only for moderate gate intensities. At higher intensities, increasingly large deviations of $\boldsymbol{j}(\tau)$ from $\boldsymbol{A}(\tau)$ become apparent before the gate field maximum (Fig. 4b) when the gate field enables transitions from CB1 (blue in Fig. 4a) to CB2 (purple). The simulations show that the wave-packet subsequently may be split up even further at other avoided crossings (CB3; shown in green).

The emergence of the population in higher conduction bands is time-delayed relative to the VUV excitation of the CB1 (Fig. 4a, c) and the delay magnitude matches the advance of the deviation in the observed LPPS signal. As an important test, when switching off the couplings between the conduction bands the time-advanced deviation disappears in the simulations. Considering the calculated group velocity difference between CB1 and CB2, even a modest population transfer from CB1 to CB2 can result in a considerable reduction of the LPPS signal, i.e., a 7.5% population transfer observed in the simulation leads to up to 17% signal reduction. Changes in the characteristics of the exciting VUV pulse in the simulation (central energy, chirp) do not qualitatively alter the picture: in addition to the contribution due to the non-parabolic dispersion relation, transfer of charge carriers to higher conduction bands leads to the largest deviation of the LPPS signal before the maximum of the gate-field envelope is reached. However, the amount of transferred charge changes with the quantitative characteristics of the exciting VUV pulse.

## Discussion

These findings emphasize that scaling optoelectronic signal processing to higher speed depends on both the fast modulation of conductivity via the rapid creation of mobile carriers (source) and the unambiguous coupling of electronic transients to external circuitry (gate). Ultrafast conductivity switching necessitates a short injection time window $\Delta\tau_{\text{FWHM}}$ and, thus, a broadband initial spread $\Delta E$ of the conduction charge carriers. The speed of excitation build-up alone is controlled by the joint density of states and may potentially involve many valence and conduction bands of the active material. However, as demonstrated here, the occupation of multiple conduction bands inhibits the robust steering of the initial Bloch wavepacket and hence compromises the capacity to map optical fields to electronic signals with high

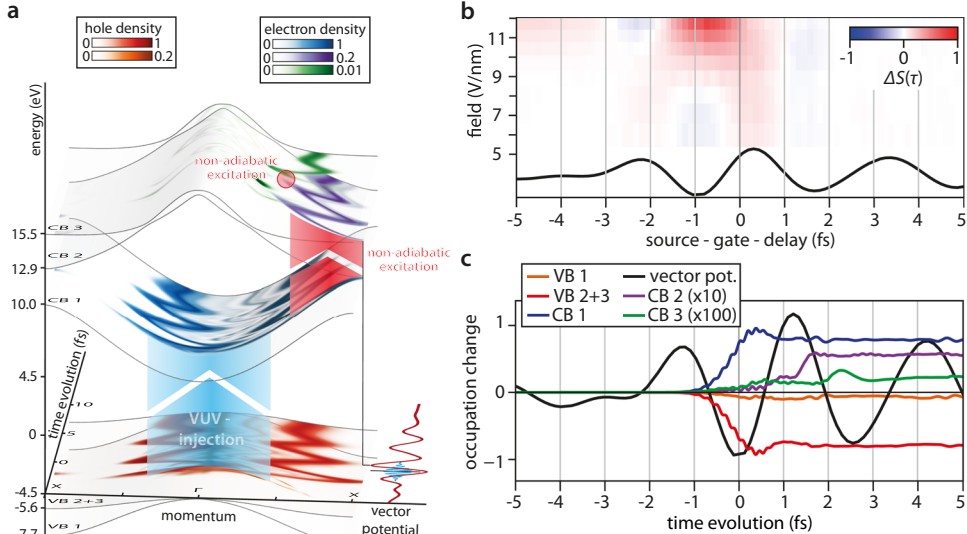

**Fig. 4 Wavepacket modeling of linear petahertz photoconductive sampling (LPPS) using optical Bloch equations along the $\Gamma-X$ direction in LiF. a** Crystal-momentum-resolved temporal evolution of the conduction band electron and valence band hole populations when the VUV source pulse (light blue, central photon energy 2.6 eV above the bandgap) precedes the maximum of the gate pulse vector potential (red) by one gate field half-cycle (source-gate-delay = −1 fs in Figs. 3b, c, 4b). The origin of the time evolution axis corresponds to the maximum of the source pulse, i.e., the moment of carrier injection into the first conduction band. The gate field strength applied in this simulation was 8.7 V/nm. Interband transitions are marked by the light blue and red areas. After the interaction of the excited carriers with the gate pulse, occupation in multiple valence and conduction bands contributes to the asymptotic current. Non-coupling bands are omitted and populations are normalized to the maximum electron population for clarity. **b** Simulated LPPS signal deviation $\Delta S(\tau)$ as a function of source-gate-delay and gate intensity for the experimental gate pulse vector potential (black). As in the experiment, the largest deviations are found in the cycle before the gate-pulse maximum. The amplitude of the intraband deviation around 0 fs source-gate-delay caused by conduction band non-parabolicity is only ~4% of the maximum deviation and is smaller than the experimental uncertainty. **c** Time-dependent occupation of the LiF valence and conduction bands (colored lines). First, CB1 population is generated by the VUV source pulse. Then, delayed population of CB2 and CB3 is induced by Landau–Zener transitions driven by the gate field. The source-gate-delay and time evolution axis origin are the same as in panel (**a**). In all panels, the gate-induced linear polarization is omitted.

fidelity. Therefore, ultimate-speed optoelectronics should avoid populating multiple conduction bands.

The shortest time interval within which high-fidelity electronic signal manipulation appears to be feasible is therefore dictated by the condition that the gate pulse must not drive transitions to higher conduction bands, as these can obscure the recorded signals beyond correction. Our experiments revealed that if the source-induced occupation does not exceed the lower half of CB1 then such transitions can, indeed, be avoided up to the 1 PHz frontier for gate field strengths up to 2 V/nm, high enough to drive well detectable currents in macroscopic circuitry. This is the major finding of our work reported here.

Assuming equal occupation of all states in $\Delta E$, we define the band-structure limit of ultrafast optoelectronics—analog to the Fourier limit of a pulse with rectangular spectral amplitude[40] (to account for the hard spectral limits imposed by the band structure)—as the product of half of the first conduction band bandwidth $\frac{\Delta E_{CB1}}{2}$ and its fastest linear response time $\Delta\tau_{FWHM}$

$$\Delta\tau_{FWHM}\frac{\Delta E_{CB1}}{2} > 3.66 \text{ eV fs}. \quad (3)$$

For LiF with $\Delta E_{CB1} = 6.2\,\text{eV}$[20] this connection yields $\Delta\tau_{FWHM} > 1.2\,\text{fs}$. This is, indeed, consistent with the excitation time of $\Delta\tau_{FWHM} \approx 1.4\,\text{fs}$ evaluated from our experiments, confirming that future solid-state optoelectronics may efficiently operate up to the 1 PHz frontier.

The optimal setting for solid-state optoelectronics thus requires materials with large bandgaps (for avoiding photodoping across the bandgap by the gate field), a large bandwidth of the first conduction band (for small switch-on times $\Delta\tau_{FWHM}$), weak coupling between CB1 and higher-lying bands and strong gate

fields on the V/nm level to assure sufficient carrier velocities in the nm/fs range. The latter point is not only important for enabling measurable signal amplitudes in our sampling setup[27,28] but also for future devices, to ensure that a created electronic signal clears the interaction zone before the arrival of a new optical signal, a minimum prerequisite for a rapid succession of switching processes. Field sampling based on tunneling ionization with a perturbation[41] has been demonstrated using field enhancement at a gold nanoantenna[42]. So far, the technique focused on a vacuum channel for electron transport, but the concept may be transferable to tunneling into the conduction band of a solid medium. In such an experiment, the sampled signal is encoded in the tunneling rate, which, according to the above discussion, does not suffer from the band-structure limit. However, as the electrons must transit the solid medium before detection, the limit dictated by Eq. (3) would still apply.

The demonstrated time- and band-resolved experimental protocol allows distinguishing interband from intraband dynamics and the controlled creation, manipulation, and observation of optical field-driven electronic signals in dielectrics coupled to external electronic circuitry. We have identified resonant photo-injection of sub-optical-cycle-duration Bloch wavepackets as an enabling concept for optical-clock-rate electronic applications.

## Methods

**Laser system and VUV generation**. An amplified and spectrally broadened titanium sapphire laser system generates carrier-envelope phase-controlled visible few-cycle laser pulses (780 nm central wavelength, 350 µJ pulse energy). We use 80% of the available power to generate VUV source pulses via frequency up-conversion in an argon gas jet. The remaining 20% are individually compressed and used as gate field.

Our setup suppresses more than 99% of the driving radiation after high-harmonic generation using an aperture and the strong divergence difference between the generated VUV and the VIS/NIR driving radiation. We verified that the remaining driving radiation does not generate photocurrents by performing a measurement with the high-harmonic gas target turned off. The streaking measurements were performed using a 0.16 µm-thick scandium foil sandwiched between two 0.04 µm-thick layers of aluminum. This combination provides a transmission window opening at 50 eV photon energy and thus allows recording well-defined streaking traces with neon as target gas.

The only filter material with a transmission window corresponding to the bandgap of LiF is indium. However, because the transmission of even thin indium foils is low in the spectral region of interest (see below) and to maximize the detected signal, we did not apply spectral filtering during the current measurements. However, the phase matching in the VUV generation process was tuned such that more than 90% of the generated VUV photons had photon energies within the indium filter transmission window:

To explore the distribution of the generated VUV in the relevant spectral region, we performed photocurrent measurements after spectrally selective indium and aluminum foils using an extreme ultraviolet photodiode (Opto Diode Corp. AXUV100). The photocurrent measured for the radiation transmitted through a 0.2 µm-thick indium filter was 2.1 times higher than after transmission through a 0.5 µm-thick aluminum filter.

When including the unavoidable oxidation of the filters, the indium transmission window opens at 11 eV photon energy and closes at 17 eV. By comparing the photon flux after one and two identical 0.2 µm indium filters, we measured an average transmission of 4.7%. Aluminum forms a self-limiting native oxide layer. When including the oxide layer on both surfaces of the filter, the transmission window of the 0.5 µm aluminum filter opens at 15 eV photon energy and transmission exceeds 15% for photon energies >21 eV. Even when we assume that all radiation detected after the aluminum filter is within the window between 17 and 25 eV, correcting for the transmission of the indium and the aluminum filters and the photodiode detection efficiency yields that more than 90% of the VUV photons in our experiments have energies within the indium transmission window between 11 and 17 eV photon energy. As the spectrum of the generated high-harmonic radiation reaches beyond 60 eV photon energy, the actual ratio of radiation within the indium window is likely higher.

**Current detection, comparison, and sample geometry**. We separate gate field-dependent signals from static background signals by reversing the gate electric field (via changing the carrier-envelope-phase by $\pi$) every second laser pulse and recording currents occurring at half the laser repetition rate via lock-in detection. We amplify and impedance-match currents before detection via transimpedance amplification (FEMTO Messtechnik GmbH, DLPCA). When averaging over an estimated $10^8$ vacuum-ultraviolet photons per second, we achieve a signal-to-noise ratio for the gate pulse intensity structure of 26 dB in the linear regime, i.e., we can detect light pulses down to intensities of 1 GW/cm². Currently, the signal-to-noise ratio is determined by the VUV source power and electrical detection noise, thus improving the VUV flux should directly increase the signal-to-noise ratio. Within the current noise level, electric field amplitude differences of >15% exceed the experimentally obtained standard deviation (SD). Evaluation of $\Delta S(\tau)$ at six amplitude maxima of the reference (timing according to Fig. 3b) yields the results in Supplementary Table 1.

The electric field of the gate laser pulses in our setup is stable over an entire measurement day due to a feed-forward stabilized carrier-envelope phase. To account for slow changes of the arm length in our interferometer, we align the reference amplitudes $S_{I_0}(\tau_0)$ to the gate intensity-dependent current amplitudes $S_I(\tau)$ in the definition of $\Delta S(\tau) = S_I(\tau) - \alpha S_{I_0}(\tau_0)$ and in Fig. 3 using a least-squares fit.

The samples employed in the studies were 1 mm-thick off-the-shelf lithium fluoride VUV windows polished for high transmission at 120 nm wavelength (Korth Kristalle GmbH) without specified crystal direction. To exclude signals caused by the interaction of gate laser pulses with the electrodes, we employed two electrode geometries with equal outcome: in sample geometry (a), we manufactured electrodes by applying silver-enriched conductive epoxy glue (EPO-TEK H22, Epoxy Technologies Inc.) directly to the sample surface. The resulting ~1 mm gap between the electrodes was large enough to exclude overlap of the near-infrared gate pulses (focused by a spherical mirror ($f = 1$ m), FWHM diameter ~ 150 µm) and the VUV source pulses (focused by a grazing incidence toroidal mirror ($f = 0.72$ m), FWHM diameter < 30 µm) with the electrodes. We amplified and detected currents for both electrodes independently. When the source and gate pulses were not centered between the electrodes, the signal strength for the closer electrode increased. However, if both pulses fully illuminated an electrode instead of the lithium fluoride, the detected signal disappeared. To completely exclude effects of the light pulses overlapping with the electrodes, sample geometry (b) used a copper-on-FR4 printed-circuit-board. The etched copper electrodes were placed on the lithium fluoride surface and the sample was illuminated from the backside of the printed circuit through a 300 µm hole. That way, illumination of the now-covered electrodes by both the gate and source pulse was excluded and current signals were still detected. For the neon LPPS measurements, we use an effusive gas nozzle located between two semi-cylindrical stainless-steel electrodes with ~4 mm

radius and ~10 mm length, such that both beams pass the electrodes comfortably. The electrodes collect photoelectrons emitted into a large solid angle around the interaction zone, which is confined to a small volume around the effusive gas nozzle by the rapidly decreasing neon density.

**Estimation of propagation effects**. Long propagation in LiF would lead to a strong group delay walk-off between the source and gate pulses and thus could introduce measurement errors. However, strong absorption of the injecting VUV light limits its penetration depth to below 15 nm and the average group delay walk-off to below 70 as. This is corroborated by the accuracy of the spectral phase and field retrieval.

**Streaking measurements**. To benchmark the gate electric field retrieved using LPPS we replace the solid-state sample with an effusive jet of neon and detect photoelectrons via a time-of-flight spectrometer oriented along the polarization direction of the gate field. The VUV radiation high energy cut-off after a thin aluminum-scandium filter provides sufficient photon flux for such a measurement without readjustment of the high-harmonic generation. The recorded spectrogram (Fig. 2c) shows clear attosecond streaking character, thus allowing extraction of the gate electric field vector potential via the streaking momentum shift of the electrons. This avoids pitfalls induced by applying the central-momentum approximation in more elaborate extraction schemes for the relatively small kinetic electron energies in this case. To scrutinize that the high-energetic radiation used in attosecond streaking is not the source of the currents in LiF, we performed the following test: a 0.5 µm-thick aluminum filter transmits more than 15% of the photons with energies larger than 20 eV. However, it attenuates the detected current amplitude in LiF by a factor of ~60. The factor of 10 mismatch between the filter and the current attenuation excludes the high-energetic radiation as the source of the current signal. The observed current/transmission ratio only matches for photon energies close to the opening of the aluminum transmission window between 15 and 17 eV.

**Extraction of the dominant injection time**. As cross-correlation between the vector potential $A_{\text{Gate}}$ of the gate field $E_{\text{Gate}}$ and the intensity envelope of the current pulse $I_{\text{UV}}$ injected by the VUV source light pulse, the measured LPPS signal $S(\tau)$, or more precisely the amplitude $\frac{|S(\omega)|}{\sqrt{|I_{\text{Gate}}(\omega)|}}$ and phase $\phi(S(\omega)) - \phi(E_{\text{Gate}}(\omega))$ of the spectral transfer function, reveal the source pulse envelope via Fourier transformation. Starting from $S(\tau)$

$$S(\tau) \propto \int_{-\infty}^{\infty} I_{\text{VUV}}(t) A_{\text{Gate}}(t - \tau) \mathrm{d}t \tag{4}$$

and its Fourier transform

$$S(\omega) \propto I_{\text{VUV}}(\omega) A_{\text{Gate}}(\omega) \tag{5}$$

we find for the frequency distribution of the VUV pulse

$$I_{\text{VUV}}(\omega) \propto \frac{S(\omega)}{A_{\text{Gate}}(\omega)} \propto \frac{i\omega\, S(\omega)}{E_{\text{Gate}}(\omega)} = i\omega \frac{|S(\omega)| e^{i\phi(S(\omega))}}{|E_{\text{Gate}}(\omega)| e^{i\phi(E_{\text{Gate}}(\omega))}}$$
$$\propto i\omega \frac{|S(\omega)|}{\sqrt{|I_{\text{Gate}}(\omega)|}} e^{i(\phi(S(\omega)) - \phi(E_{\text{Gate}}(\omega)))}. \tag{6}$$

Thus, we can derive the temporal intensity envelope of the VUV pulse or, equivalently, the excitation probability by the source pulse,

$$I_{\text{VUV}}(t) \propto \text{IFT}\left( i\omega \frac{|S(\omega)|}{\sqrt{|I_{\text{Gate}}(\omega)|}} e^{i(\phi(S(\omega)) - \phi(E_{\text{Gate}}(\omega)))} \right) \tag{7}$$

giving access to the FWHM duration of the excitation process $\Delta\tau_{\text{FWHM}}$.

As the method employs the rapidly oscillating electric field of the gate pulse, it offers significantly higher temporal resolution as compared to using the envelope of the gate pulse.

The spectral amplitude recorded by attosecond streaking is affected by a finite photoionization time similarly to the LPPS measurement. A detailed retrieval of spectral features is not possible for the delay ranges attainable in attosecond streaking (see Supplementary Fig. 1). Therefore, we also show the spectral amplitude retrieved from an LPPS scan with an extended delay range and use a calibrated grating spectrometer to determine the spectral response (see Fig. 2c and Supplementary Fig. 1).

**Simulations employing the Bloch equations**. The aim of the simulations is an understanding of the signatures of interband and intraband transfer of the source excited carriers by the gate signal. By using the density functional theory-derived energies and accurate coupling matrix elements, we ensure that the complex interband transfer is captured correctly. However, a quantitative comparison with the experiment requires sampling the full Brillouin zone. To simulate the field-driven dynamics, we solve the equation of motion for the one-particle reduced density matrix in the basis of Houston orbitals. We calculate the LiF ground state in

the local-density approximation (LDA) using the Vienna ab-initio simulation package (VASP). As the bandgap calculated by density functional theory ($E_{Gap}$ = 8.9 eV) differs from the experimental value, we reduce the central frequency of the source pulse accordingly ($\omega$ = 11.5 eV). To capture the symmetry of the crystal we solve the Bloch equations along four one-dimensional cuts close to the $\Gamma$–$X$ direction on a dense $k$-mesh. The band structure and matrix elements of the dipole transition $p_{nm}^k$ and eigenenergies $\varepsilon_{nk}$ are extracted from the density functional theory calculation carefully enforcing continuity of $p_{nm}^k$. We include four valence and eight conduction bands in the simulation. All simulations were converged with respect to the number of $k$-points and the time-step.

Linearity of the LPPS signal can be maintained for high gate fields by operating on electrons in the ionization continuum resembling a single parabolic conduction band. We model neon with the same simulation representing the 2$p$-state by a flat valence band VB. Ionization proceeds to a single parabolic conduction band CB with the bandgap set to the ionization potential of neon $I_P$ = 21.6 eV, coupled by $p_{CBVB}^k \propto \frac{1}{\varepsilon_{CBk} - \varepsilon_{VBk}}$. The IR gate pulse induces virtual excitations due to linear polarization (AC Stark shift) which we subtracted prior to plotting the results in Fig. 4 in the main text. As in the experiment, we find a linear mapping between vector potential and resulting LPPS signal at all intensities for neon, where electrons experience a perfectly parabolic single-band free-electron dispersion relation in the ionization continuum and coupling to higher bands does not exist.

**LPPS with a chirped VUV source pulse**. In the simulations, the timing and efficiency of the interband transitions can be further investigated by considering chirped source pulses. Translating the high-frequency spectral portion under the envelope of the VUV pulse injects carriers at $k$-values closer to the avoided crossings, thus the largest deviation $\Delta S(\tau)$ appears earlier within the optical half-cycle preceding the gate field maximum. Supplementary Fig. 2 explores this effect for high field intensities and large source pulse chirp (0.27 fs$^2$) to highlight this behavior. We consider this mechanism as the source of the small time-advance of the maximal deviation observed in the experiment versus the simulations.

## Data availability

The data that support the findings of this study are available at https://doi.org/10.6084/m9.figshare.18675887 and from the corresponding author upon request.

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

## Acknowledgements

This research is based upon work supported by the US Air Force Office of Scientific Research under award number FA9550-16-1-0073 (M.S., N.K., F.K.). Calculations were performed on the Vienna Scientific Cluster (VSC) and the Oakforest-PACS super-computer of the University of Tokyo in Japan. M.O. acknowledges a Feodor-Lynen Fellowship from the Alexander von Humboldt Foundation. Furthermore, we acknowledge funding by the Austrian Science Fund FWF under project numbers SFB F41 (ViCoM) (F.L., J.P.B., V.S.) and W1243-N16 (Solids4Fun) (I.F., J.P.B.), doctoral college TU-D of TU Wien (V.S.), and by the Max-Planck-Gesellschaft within IMPRS-APS (V.S., I.F., S.D.).

## Author contributions

M.O., M.S., F.K. developed the concept and wrote the manuscript. M.O., K.G., K.S., L.L., F.S., J.P.B., D.Z., J.A.G., M.W., N.K., M.S. performed the experiment and analyzed experimental data. I.F., V.S., S.D., C.L., F.L., J.P.B. performed the theoretical modeling and wrote the manuscript.

## Funding

## Competing interests
The authors declare no competing interests.
