## [Peer Review File · Nature Communications]

The speed limit of optoelectronicsEditorial Note: This manuscript has been previously reviewed at another journal that is not operating a transparent peer review scheme. This document only contains reviewer comments and rebuttal letters for versions considered at *Nature Communications*.

REVIEWERS' COMMENTS

Reviewer #2 (Remarks to the Author):

Ossiander et al. have revised their manuscript in response to the second round of reviewer comments when it was considered for [another journal – redacted]. I myself have not brought up new points in the second round and also do not do it here, hence I maintain my positive view of the work.

In their newly revised version, the authors have done a good job at addressing the criticism of the other reviewers. In response to Reviewer 3, they have introduced a more complete and satisfying statistical analysis of the experimental data and have argued correctly that the method of attosecond streaking for retrieving the attochirp is not suitable for the photon energy of the incoming VUV light pulse. Reviewer 4 compared the work of the authors to vacuum gap experiments (Bionta et al., Nat. Photon. 2021, appeared much after the initial submission of this manuscript) and found it to some degree inferior. However, I believe that such a comparison is not fair due to the fact that the manuscript involves a wide bandgap dielectric with a complex band structure rather than plain vacuum plus nano-electrodes. This manuscript highlights the potential and the challenges of fully solid-state PHz optoelectronics and deserves wide attention. As field samplers, both directions seem very promising, but field sampling is not at the heart of this manuscript. The comment by Reviewer 5 is also addressed properly, discussing the magnitude of higher conduction band populations and its correlation with the measured signal.

In the light of my previous review comments and the authors' excellent response to the remaining criticism, I am strongly recommending it for publication in *Nature Communications* in its present form.

Reviewer #3 (Remarks to the Author):

After consideration of the comments and answers to reviewers and of the revised manuscript and supplementary material, the manuscript can be published as it is in *Nature Commun.*

Reviewer #4 (Remarks to the Author):

After reading the revised manuscript, I am in general quite satisfied. The manuscript is very strong, and the work very timely. I support its publication in Nature Communications.

There is only one remaining item that I feel strongly should be revised, and that is the title. Currently it reads "The speed limit of optoelectronics". I feel this title is too general and a bit misleading as they rather study the speed limit of optoelectronics with *solid-state* channels. They clearly note this throughout the text correctly.

As the authors point out, direct tunneling-rate field-sampling methods, such as TIP-TOE, can be performed in nanoscale systems having vacuum/air gaps, and that in this configuration their findings would not suffer the limitations that they describe in this work. I do, however, agree completely with the authors in their revised manuscript and referee reply that if such TIP-TOE-like measurements were performed in a solid state system, they would suffer the exact same limitations as described in this work. These results are indeed illuminating as to how far solid state systems can be pushed. However, there is no reason a device cannot utilize nanoscale vacuum channels for transport and still be considered an optoelectronic device. If anything, this work indicates that vacuum channels in optoelectronics are a necessity to push such methods to yet higher frequencies, probably then ultimately limited by physical size limitations, signal propagation, and readout.

In short, while very important, I don't believe this work offers a definitive speed limit of optoelectronics in the most general context as the title implies.

//NOTE: The following note is a followup discussion to reply to a point made in the referee response. For the sake of scientific discussion, I am replying to it here. However, I realize this is not the main point of this manuscript, and should not impact the decision for publication of this work. //

Regarding the comment in the reply about the $1/\omega$ impact in streaking-like measurements, I stand by the fact that this has a more profound impact as it is fundamental -- i.e. it cannot be avoided with material choice or design. Furthermore, it affects the amplitude of the signal, limiting the overall SNR. The phase impact pointed out in Bionta et al. is due to the nanoantenna itself, and has the potential to be avoided via design or material choice -- and by virtue of being a phase artefact, can simply be accounted for with proper calibration without impacting the fundamental SNR.

Reviewer #5 (Remarks to the Author):

Review Comment to authors:

I twice reviewed the authors' manuscript which was submitted to [another journal – redacted] a few months ago. This manuscript by M. Ossiander et al., reports a demonstration of optical field sampling by using a wide-gap dielectric, lithium-fluoride, via linear absorption of ultraviolet high-order harmonic pulses with near-1-fs duration, which is named as linear petahertz photoconductive sampling (LPPS). As a similar technique, there is a series of previous reports based the nonlinear photoconductive sampling (NPS) scheme. The most important scientific value of their work is to clearly show the fidelity of the measured electric field by using VUV single-photon excitation into the first conduction band as uniformly as possible and comparing the resultant sampling electric field with the most established attosecond streaking measurement. This careful comparison revealed a clear deviation of the LPPS signal from the reference signal (the attosecond streaking measurement), and leading to the conclusion that the main contribution which limits the fidelity is most likely to the transition from one conduction band to higher energy bands.

In previous reports based on NPS scheme, the fidelity of the measured electric field has not been discussed in detail. All NPS previous reports just showed the oscillatory current signal, whose periodicity nearly corresponds to the input optical pulse field. There was no discussion about how the obtained signal reproduce the input optical field. The work by Ossiander et al. digs into this point by using single-photon excitation with VUV instead of strong-field excitation with NIR. As a matter of fact, there is no clear observation for how electrons populate in conduction bands via strong-field excitation in wide-gap materials so far. As the authors describe, there is a possibility that strong-field excitation populates several conduction bands and reduces the fidelity of the measurement originating from a reduced effective bandwidth.

By populating electrons into the first conduction band as uniformly and selectively as possible with 1-fs VUV linear excitation, the authors successfully changed the situation from the previous NPS experiments. By simultaneously measuring the vector potential of the gate pulse by the well-established attosecond streaking measurement, they found a clear deviation of the LPPS signal from the reference signal measured at relatively lower intensity. This finding is the most important value of this work in my opinion.

According to the authors' consideration, the fidelity measured by previous NPS schemes is not good because strong-field excitation induces multi-band population bands. This effect strongly depends on the band structure in the conduction band and the band gap of wide-gap materials as well as the intensity of the injection pulse. Ideally, they should demonstrate an experimental difference between

the LPPS and the NPS schemes and comprehensive investigation of the fidelity by measuring a variety of wide-gap materials. However, this will be our promising future works.

As the authors described, the development of ultimate-speed electron-based signal processing technology operated by light waveform in optoelectronic circuitry is currently one of the most important and broad-interest topics in the future petahertz technology. Therefore, I well agree that the topic described in the current manuscript is in the scope of Nature Communications. With the authors' revision at this transfer to NC, the present manuscript describes their scientific important achievement and their novelty which differs from a series of previous NPS schemes based on strong-field excitation in a clear and straightforward manner. In addition, the authors' comment on my another concern about the reason why small or modest population transfer from CB1 to other higher energy bands affects the signal distortion of $\Phi S(\omega)$ as shown in Fig. 3 (b) and (c) by showing the estimation of group velocity change induced by the population transfer quantitatively in their response letter.

In summary, from the above consideration, I feel that the present version satisfied the criteria for publication in Nature Communications.

Specific comments and answers to the reviewers

In the following, we give a one-to-one response (in black) to the individual reviewer's comments which are printed in *grey italic*.

Reviewer #2 (Remarks to the Author):

Ossiander et al. have revised their manuscript in response to the second round of reviewer comments when it was considered for [another journal – redacted]. I myself have not brought up new points in the second round and also do not do it here, hence I maintain my positive view of the work. In their newly revised version, the authors have done a good job at addressing the criticism of the other reviewers. In response to Reviewer 3, they have introduced a more complete and satisfying statistical analysis of the experimental data and have argued correctly that the method of attosecond streaking for retrieving the attochirp is not suitable for the photon energy of the incoming VUV light pulse. Reviewer 4 compared the work of the authors to vacuum gap experiments (Bionta et al., Nat. Photon. 2021, appeared much after the initial submission of this manuscript) and found it to some degree inferior. However, I believe that such a comparison is not fair due to the fact that the manuscript involves a wide bandgap dielectric with a complex band structure rather than plain vacuum plus nano-electrodes. This manuscript highlights the potential and the challenges of fully solid-state PHz optoelectronics and deserves wide attention. As field samplers, both directions seem very promising, but field sampling is not at the heart of this manuscript. The comment by Reviewer 5 is also addressed properly, discussing the magnitude of higher conduction band populations and its correlation with the measured signal. In the light of my previous review comments and the authors' excellent response to the remaining criticism, I am strongly recommending it for publication in Nature Communications in its present form.

We thank the reviewer again for her/his very positive assessment of our manuscript and responses.

Reviewer #3 (Remarks to the Author):

After consideration of the comments and answers to reviewers and of the revised manuscript and supplementary material, the manuscript can be published as it is in Nature Commun.

We thank the reviewer for her/his recommendation for publication.

Reviewer #4 (Remarks to the Author):

After reading the revised manuscript, I am in general quite satisfied. The manuscript is very strong, and the work very timely. I support its publication in Nature Communications.

We thank the reviewer for her/his positive assessment and supporting publication.

*There is only one remaining item that I feel strongly should be revised, and that is the title. Currently it reads "The speed limit of optoelectronics". I feel this title is too general and a bit misleading as they rather study the speed limit of optoelectronics with *solid-state* channels. They clearly note this throughout the text correctly.*

As the authors point out, direct tunneling-rate field-sampling methods, such as TIP-TOE, can be performed in nanoscale systems having vacuum/air gaps, and that in this configuration their findings would not suffer the limitations that they describe in this work. I do, however, agree completely with the authors in their revised manuscript and referee reply that if such TIP-TOE-like measurements were performed in a solid state system, they would suffer the exact same limitations as described in this work. These results are indeed illuminating as to how far solid state systems can be pushed. However, there is no reason a device cannot utilize nanoscale vacuum channels for transport and still be considered an optoelectronic device. If anything, this work indicates that vacuum channels in optoelectronics are a necessity to push such methods to yet higher frequencies, probably then ultimately limited by physical size limitations, signal propagation, and readout.

In short, while very important, I don't believe this work offers a definitive speed limit of optoelectronics in the most general context as the title implies.

We discussed the reviewer's suggestion and concluded the concise title summarizes our work correctly due to the following reasons: with its current title, the manuscript conveys what the reviewer states here – that using vacuum channels is beneficial – to a broader readership. The manuscript highlights this by comparing the neon and lithium fluoride measurements and showing how using neon and a vacuum channel is a way to beat the discussed limit. Together with the added reference to Bionta et al. and the added discussion of how to apply or defeat the limit in such structures, we believe the manuscript presents both the solid-state and vacuum cases.

//NOTE: The following note is a followup discussion to reply to a point made in the referee response. For the sake of scientific discussion, I am replying to it here. However, I realize this is not the main point of this manuscript, and should not impact the decision for publication of this work. //
Regarding the comment in the reply about the $1/\omega$ impact in streaking-like measurements, I stand by the fact that this has a more profound impact as it is fundamental -- i.e. it cannot be avoided with material choice or design. Furthermore, it affects the amplitude of the signal, limiting the overall SNR. The phase impact pointed out in Bionta et al. is due to the nanoantenna itself, and has the potential to be avoided via design or material choice -- and by virtue of being a phase artefact, can simply be accounted for with proper calibration without impacting the fundamental SNR.

We agree that sampling the vector potential in streaking-type measurements is fundamental for unstructured materials and this has implications for certain applications. We also believe the straightforward analytical correction has benefits in other applications. We see Bionta et al.'s work as an impressive way to sample fields on the nanoscale. Still, we believe a phase shift is fundamental when using a resonant structure and therefore unavoidable. We therefore want to re-state that we see both techniques as valid and complementary pathways towards a future field sampler.

Reviewer #5 (Remarks to the Author):

I twice reviewed the authors' manuscript which was submitted to [another journal – redacted] a few months ago. This manuscript by M. Ossiander et al., reports a demonstration of optical field sampling by using a wide-gap dielectric, lithium-fluoride, via linear absorption of ultraviolet high-order harmonic pulses with near-1-fs duration, which is named as linear petahertz photoconductive sampling (LPPS). As a similar technique, there is a series of previous reports based the nonlinear photoconductive sampling (NPS) scheme. The most important scientific value of their work is to clearly show the fidelity of the measured electric field by using VUV single-photon excitation into the first conduction band as uniformly as possible and comparing the resultant sampling electric field with the most established attosecond streaking measurement. This careful comparison revealed a clear deviation of the LPPS signal from the reference signal (the attosecond streaking measurement), and leading to the conclusion that the main contribution which limits the fidelity is most likely to the transition from one conduction band to higher energy bands.

In previous reports based on NPS scheme, the fidelity of the measured electric field has not been discussed in detail. All NPS previous reports just showed the oscillatory current signal, whose periodicity nearly corresponds to the input optical pulse field. There was no discussion about how the obtained signal reproduce the input optical field. The work by Ossiander et al. digs into this point by using single-photon excitation with VUV instead of strong-field excitation with NIR. As a matter of fact, there is no clear observation for how electrons populate in conduction bands via strong-field excitation in wide-gap materials so far. As the authors describe, there is a possibility that strong-field excitation populates several conduction bands and reduces the fidelity of the measurement originating from a reduced effective bandwidth.

By populating electrons into the first conduction band as uniformly and selectively as possible with 1-fs VUV linear excitation, the authors successfully changed the situation from the previous NPS experiments. By simultaneously measuring the vector potential of the gate pulse by the well-established attosecond streaking measurement, they found a clear deviation of the LPPS signal from the reference signal measured at relatively lower intensity. This finding is the most important value of this work in my opinion.

According to the authors' consideration, the fidelity measured by previous NPS schemes is not good because strong-field excitation induces multi-band population bands. This effect strongly depends on the band structure in the conduction band and the band gap of wide-gap materials as well as the intensity of the injection pulse. Ideally, they should demonstrate an experimental difference between the LPPS and the NPS schemes and comprehensive investigation of the fidelity by measuring a variety of wide-gap materials. However, this will be our promising future works.

As the authors described, the development of ultimate-speed electron-based signal processing technology operated by light waveform in optoelectronic circuitry is currently one of the most important and broad-interest topics in the future petahertz technology. Therefore, I well agree that the topic described in the current manuscript is in the scope of Nature Communications. With the authors' revision at this transfer to NC, the present manuscript describes their scientific important achievement and their novelty which differs from a series of previous NPS schemes based on strong-field excitation in a clear and straightforward manner. In addition, the authors' comment on my another concern about the reason why small or modest population transfer from CB1 to other higher energy bands affects the signal distortion of $\Delta S(\tau)$ as shown in Fig. 3 (b) and (c) by showing the estimation of group velocity change induced by the population transfer quantitatively in their response letter. In summary, from the above consideration, I feel that the present version satisfied the criteria for publication in Nature Communications.

We thank the reviewer for her/his detailed considerations and the recommendation for publication.